# Clinical and Genomic Characterization of Carbapenem-Resistant *Klebsiella pneumoniae* with Concurrent Production of NDM and OXA-48-like Carbapenemases in Southern California, 2016–2022

**DOI:** 10.3390/microorganisms11071717

**Published:** 2023-06-30

**Authors:** Stacey Cerón, Zackary Salem-Bango, Deisy A. Contreras, Elizabeth L. Ranson, Shangxin Yang

**Affiliations:** 1Department of Pathology and Laboratory Medicine, UCLA David Geffen School of Medicine, Los Angeles, CA 90095, USA; sceron@mednet.ucla.edu (S.C.); zbango@mednet.ucla.edu (Z.S.-B.); 2Department of Pathology and Laboratory Medicine, Cedars-Sinai Medical Center, Los Angeles, CA 90048, USA; contrerasd@csmc.edu; 3Division of Infectious Diseases, UCLA David Geffen School of Medicine, Los Angeles, CA 90095, USA; eranson@mednet.ucla.edu; 4West Los Angeles VA Medical Center, Los Angeles, CA 90073, USA

**Keywords:** carbapenem-resistant *Klebsiella pneumoniae*, NDM, OXA-181, OXA-232, dual carbapenemase producer, whole-genome sequencing, antimicrobial resistance, Southern California

## Abstract

The global emergence of carbapenem-resistant *Klebsiella pneumoniae* (CRKP) has become a critical public healthcare concern due to treatment challenges and high mortality. In recent years, there has been an increase in cases of CRKP co-producing New Delhi metallo-β-lactamases (NDM) and oxacillinase 48 (OXA-48)-like carbapenemases in the US. The aim of this study was to correlate the clinical and genomic characteristics of CRKP co-producing NDM and OXA-48-like carbapenemases isolated from patients in Southern California since 2016. Whole-genome sequencing was performed on clinical isolates obtained from various sources, including blood, abdominal fluid, wounds, and urine. Genetic diversity was observed in these CRKP, including ST-14, ST-16, ST-167, ST-437, ST-2096, and ST-2497 lineages. Phylogenetic analysis revealed two closely related clusters (ST-14 and ST-2497), with single nucleotide polymorphism (SNP) differences ranging from 0 to 36, suggesting a possible local spread of these CRKP. Significant antimicrobial resistance (AMR) genes were identified in these CRKP, including *bla_NDM-1_*, *bla_NDM-5_*, *bla_OXA-232_*, *bla_OXA-181_*, *bla_CTX-M-15_*, *armA*, *tet(A)*, and *tet(D).* Moreover, *pColKP3*-type and *Inc*-type plasmids known to harbor AMR genes were also detected in these isolates. Most of the patients infected with this rare type of CRKP died, although their severe comorbidities also played important roles in their demise. Our study highlighted the extremely limited treatment options and poor clinical outcomes associated with these dual-carbapenemase-producing CRKP. Real-time genomic surveillance of these unusual and deadly CRKP can provide critical information for infection prevention and treatment guidance.

## 1. Introduction

The members of *Enterobacteriaceae* are Gram-negative bacteria that can cause community- and healthcare-acquired infections (HAIs) in the bloodstream, respiratory tract, and urinary tract [1,2]. In a recent study, *Enterobacteriaceae* were found to be the cause of 23–31% of HAIs in patients that were in the adult, pediatric, and oncology wards of acute care hospitals in the United States [3]. Notably, *Escherichia coli*, *Klebsiella*, and *Enterobacter* species were the most frequent pathogens associated with HAIs [4]. Formerly, *Enterobacteriaceae* infections were treated with cephalosporins; however, as antibiotic resistance due to extended-spectrum β-lactamase (ESBL) production became more prevalent, their use fell out of favor [1,5]. Carbapenems, a β-lactam antibiotic, are currently used as part of the last line of defense against drug-resistant Enterobacteriaceae [6,7]. However, the significant use of carbapenems and other factors, such as travel, resulted in the emergence of carbapenem-resistant *Enterobacteriaceae* (CRE) [8,9].

Resistance to carbapenems arises from carbapenemase production (carbapenemase-producing CRE) and/or changes to the membrane proteins/production of other β-lactamases (non-carbapenemase-producing CRE) [10]. Carbapenemase-producing CRE (CP-CRE) can be further classified into Ambler Class A, B, and D based on sequencing motifs [11,12]. *Klebsiella pneumoniae* carbapenemases (KPC), members of Ambler Class A, hydrolyze carbapenems, thus allowing bacterial cell wall synthesis [13]. Ambler Class B carbapenemases are also able to hydrolyze β-lactams but do so by using zinc and not serine in their active sites. New Delhi metallo-β-lactamase (NDM), Imipenemase metallo-β-lactamase (IMP), and Verona integron-encoded metallo-β-lactamase (VIM) are examples of Ambler Class B members [14]. Lastly, Oxacillin (OXA) carbapenemases are members of the Ambler Class D, which hydrolyze carbapenems in the similar way as those members belonging to Ambler Class A [15]. The acquisition of methods of resistance (mainly through plasmid-based horizontal gene transfer) and the global distributions are also distinguishing characteristics of these carbapenemases [10,16].

Due to this myriad of resistance mechanisms, CRE infections pose a unique and worrisome challenge for infectious disease practitioners to treat. Moreover, carbapenem resistance is often associated with resistance to multiple other antimicrobial agents, further complicating an already challenging situation [17]. Certain reagents are often utilized on an empiric basis or after susceptibility testing, such as ceftazidime-avibactam, amikacin, gentamicin, fosfomycin, tigecycline, polymyxin B, colistin, cefiderocol, and plazomicin. Their use depends on the provider’s experience, the local antibiogram, availability, and the toxicity of specific agents. However, the evidence to support selection of one agent versus another is minimal due to limited randomized controlled trials [18]. Even with the early identification of CRE and appropriate therapy, treatment failure remains quite high, with some studies estimating mortality near to 30% or even higher in patients with CRE pneumonia [19,20,21].

In the United States, the spread of the *Klebsiella pneumoniae* carbapenemase (KPC) led to an increase in the prevalence of CRE in the 1990s [5]. Other carbapenemases have also been reported in patients traveling to endemic regions [22,23]. Recently, there has been an emergence of carbapenem-resistant *K. pneumoniae* (CRKP) co-producing NDM and OXA-48-like carbapenems, especially in the Arabian Peninsula [24,25]. Given the complexity of treating CRE infections, dual- or multi-carbapenemase-producing CRE pose a significant public health threat due to limited therapeutic options [26,27]. In this study, we analyzed the resistance profiles of CRKP with the concurrent production of NDM and OXA-48-like carbapenemase isolates obtained from six patients in Southern California from 2016 to 2022.

## 2. Material and Methods

### 2.1. Identification of CRKP Isolates with NDM and OXA-48-like Carbapenemase Co-Production

Clinical CRKP isolates were worked up in the UCLA Clinical Microbiology Laboratory, from 4 patients admitted to UCLA Medical Center, and 2 patients from outside hospitals. The specimens included blood, hematoma, penile, peritoneal cavity fluid, expectorated sputum, tissue (abdominal and parotid), and urine. Table 1 provides a summary of the demographics and clinical presentation of the cohort. Routine identification (culture and MALDI-TOF) and broth microdilution-based susceptibility testing identified CRKP, and Xpert Carba-R (Cepheid, Sunnyvale, CA, USA) detected the co-production of NDM-1 and OXA-48-like carbapenemase.

### 2.2. Whole-Genome Sequencing and Genomic Analysis

Genomic DNA was extracted from the clinical isolates using the EZ1 Blood and Tissue Kit and the EZ1 Advanced XL instrument (Qiagen, Hilden, Germany). The Qubit 1xdsDNA HS assay was used to quantify the extracted DNA via the Qubit 3.0 Fluorometer (ThermoFisher Scientific, Waltham, MA, USA). The sequencing library was created using the Nextera DNA Flex Library Prep Kit (Illumina, San Diego, CA, USA) based on the manufacturer’s instructions. Sequencing was performed using MiSeq (Illumina, San Diego, CA, USA) with a 2 × 300 bp v3 protocol. The metrics for sequencing quality control included having a Q30 score of ≥85% and a cluster PF of ≥60%. At least 1 million reads were acquired for each isolate. AMR gene profiles and core genome MLST (cgMLST) were analyzed using the 1928 Diagnostics (Gothenburg, Sweden) cloud-based analytical service [28,29]. Single nucleotide polymorphism (SNP) phylogenetic analysis was performed using the “Basic Variant Detection” tool in CLC Bio Genomic Workbench (Qiagen, Germantown, MD, USA). This tool was used to create a variant track following the read mapping to the reference genome. Default settings were used for various sequence quality control parameters, including ≥85% allele frequency and ≥10X minimum coverage/position for variant calling. The variant tracks for each isolate were then compared using the “Create SNP Tree” tool. SNP trees were created using maximum likelihood algorithms. The sequence reads were assembled using the CLC bio de novo assembly tool with the default setting. The plasmid types were identified by the Center of Genomic Epidemiology (Lyngby, Denmark). The sequence data were deposited in the NCBI Genbank (BioProject ID: PRJNA986585).

### 2.3. Chart Review and Research Ethics

The patient’s clinical history was retrospectively reviewed. This study was reviewed by the UCLA Human Research Protection Program and received an institutional review board exemption.

## 3. Results

### 3.1. Clinical Characteristics

Our limited cohort consisted of six patients from 2016 to 2022, with distinct medical histories and clinical manifestations. Four were directly treated at our hospital facility, whereas two received treatments at outside centers but utilized our laboratory facility for assistance in susceptibility testing and genetic analysis. Five out of the six of our cohort died during their respective hospitalizations. The patient information of five out of the six patients was available for complete chart review, and their detailed clinical histories are described here.

### 3.2. Case 1 (2016)

A 75-year-old male who received treatment at an outside facility for a urinary tract infection. Our laboratory facility was asked to assist in susceptibility testing after his urine culture returned positive for CRE *K. pneumoniae*. Unfortunately, no additional history was available for inclusion due to the patient being located outside our treatment center. He did unfortunately pass away during this hospitalization.

### 3.3. Case 2 (2017)

A 74-year-old male originally from India with a history of ischemic cardiomyopathy, chronic kidney disease, and non-insulin-dependent type II diabetes mellitus who presented to our facility from an outside hospital for management of complex cardiogenic shock and incidental bacterial parotitis. His history was significant due to frequent travel between India and the United States. Three months prior to admission to our facility, he was noted to be in relatively stable condition by his family and outside of chronic diabetes mellitus management. However, he experienced two bouts of acute decompensated heart failure while in India and required hospitalization. One admission did require treatment in an intensive care unit equivalent in India. A month prior to the admission at our facility, he returned from India and was subsequently admitted at a facility in Central California. He was medically optimized and discharged with a wearable defibrillator. One week prior to admission at our facility, he was readmitted to the outside hospital in cardiogenic shock and was found to have a cardiorenal acute kidney injury (AKI). He was started on inotropic agents at this time and an intra-aortic balloon pump (IABP) was placed, at which point he was transferred to our facility for a higher level of care.

Upon arrival, he was found to have an elevated white blood cell count (WBC), which triggered an infectious workup and empiric therapy with piperacillin-tazobactam. Of note, he had had an ongoing left-sided cheek mass for three years with some associated numbness that had not been evaluated until admission at our facility. He denied any history of head or neck cancer or past infections. Two days after admission, respiratory cultures were found to be positive for CRKP. Fine needle aspiration (FNA) of the left parotid gland was performed and returned positive two days later for CRKP again. Blood and urine cultures remained negative at this time; thus, empiric antibiotic therapy was discontinued.

Four days after admission, he underwent percutaneous coronary intervention (PCI). At day 5 post-admission, he developed bloody bowel movements. Given the ongoing concern regarding a systemic infection secondary to the patient’s own normal flora, the decision was made to initiate therapy with polymyxin B and meropenem, with the latter being renally dosed due to ongoing AKI. On day 7 post-admission, the bleed was determined to be secondary to a duodenal dieulafoy lesion, which was treated with electrocautery and epinephrine. The following day, IABP was discontinued due to hemodynamic stability. However, over the next 24 h the patient steadily decompensated with declining venous oxygen levels, hypotension, and respiratory rates at 30 s and 40 s. At that time, it was unclear if this acute change was secondary to septic (patient was found to have rising WBC count) vs. cardiogenic shock; so, the IABP was replaced with increased inotropic and vasopressor support. There was no improvement in hemodynamics after these interventions. Ten days after admission to our facility, the patient went into pulseless electrical activity (PEA) and expired.

### 3.4. Case 3 (2018)

A 68-year-old female with a past medical history that was significant due to breast cancer (status post-lumpectomy, stable, on tamoxifen for five years), stable meningioma, type II diabetes mellitus complicated by right-sided retinal detachment, and end-stage renal disease, on hemodialysis for 8 years prior to admission, who originally presented to our facility for a dead-donor renal transplant. Prior to this admission, the patient had a remote history of a methicillin-resistant *Staphylococcus aureus* infection of her arteriovenous fistula, but no other infection-related complications; she had not traveled in over a year. The donor graft was sourced from Nevada and had no prior travel history within the previous year. No CRE cases had been sourced at the originating facility in 2.5 years.

The post-transplant period was complicated by sudden pulseless electrical activity of an unknown etiology with the return of spontaneous circulation (ROSC), though the patient had been started on ECMO and continuous veno-venous hemodialysis. Two days post-transplant, the patient developed an allograft hematoma and required an exploratory laparotomy for removal. Fourteen days post-transplant, the patient remained in critical condition, anuric, and on multiple vasopressors. At this time, the culture of her peritoneal dialysis drain and her blood cultures returned positive for CRKP, though she remained afebrile. She was started on ceftazidime-avibactam, aztreonam, and daptomycin at this time. She developed a fever over the next day, with subsequent augmentation of therapy with plazomicin. The renal allograft was removed 16 days post-admission, with no evidence of gross graft infection, but some poor wound healing. A recurrent hematoma developed, requiring repeat washout 21 days post-admission. Investigational antibiotic cefiderocol was started at this time. The patient developed a persistent gastrointestinal (GI) bleed 23 days post-admission, with a colonoscopy showing diffuse colitis and severe ulcerations; in addition, there was a concern regarding ischemic vs. infectious etiology. Due to ongoing pressor requirements, continuous renal replacement therapy, and newfound feculent output in one of her Jackson–Pratt (JP) drains, the patient’s family decided to transition to comfort care. She expired the following day.

### 3.5. Case 4 (2019)

An 89-year-old male with history of coronary artery disease (CAD) and coccidiomycosis who was transferred to our facility from India one month after cardiac arrest with ROSC, with cervical spinal cord compression/hyperextension injury secondary to a fall during the arrest. He was originally managed conservatively at a hospital in India but deteriorated and required intubation and eventual decompression and fixation of C3-4 with tracheostomy as well. His hospitalization in India was complicated by *Staphylococcus haemolyticus* and *Enterococcus faecium* central venous catheter (CVC)-associated bacteremia, as well as ventilator-associated pneumonia (VAP), with sputum cultures positive for CRKP, *Acinetobacter baumanii* and *Aspergillus flavus.* His CVC was removed at that facility and replaced with a peripherally inserted central catheter (PICC). At the time of arrival at our facility, his antimicrobial regimen included cefepime/sulbactam, tigecycline, inhaled colistin, and voriconazole, and his PICC was replaced with a femoral CVC.

Unfortunately, one day after arrival, his course was further complicated by *Elizabethkingae* bacteremia, and he was started on minocycline. He remained relatively stable on pressors and with an ongoing ventilator requirement for the following week, with the eventual removal of his CVC due to a concern regarding central line-associated bacteremia. Two weeks after his admission, he developed increased tracheal secretions, which were found to be positive for *Candida parapsilosis*, and he was started on caspofungin. At that time, he also developed purulent discharge from his urethra; the culture of this discharge returned positive for CRKP; CKRP was also detected in the patient’s blood three days later. He did improve over the next two weeks with ongoing antibiotic therapy and continuous renal replacement therapy (CRRT), though he continued to have evidence of systemic bacterial infection. However, he spontaneously developed severe right upper lung mucus plugging and atelectasis 1.5 months post-admission, with worsening renal failure and increasing ventilator requirements with ongoing severe leukocytosis despite constantly updated antibiotic regimens. Due to no improvement with ongoing maximal medical therapy, the decision was made by the family to palliate the patient, and he expired two months after his transfer to our facility.

### 3.6. Case 5 (2020)

A 64-year-old male with a past medical history that was significant due to chronic obstructive pulmonary disease (COPD), pulmonary hypertension, CAD, diverticulitis, recurrent pneumothorax with bronchopleural fistula, history of hepatitis C (negative status post-therapy), and latent tuberculosis with a distant history of approximately 6 months of treatment; regarding his status, he had a bilateral lung transplant 3 months prior to admission; he presented to the emergency department with new onset hemoptysis and pleuritic chest pain. The patient was born and raised in New York but had moved to California nearly thirty years previously. He had a history of intravenous (IV) drug use but had stopped using all drugs 20 years prior to admission. He had previously been diagnosed with tuberculosis (TB) in the 1980s and endorsed treatment at that time. Since his transplant, he had been on chronic immunosuppressive therapy with tacrolimus and prednisone.

At the time of admission, he was incidentally found to be positive for COVID-19. He was started on vancomycin and piperacillin-tazobactam empirically in the emergency department. Due to a concern regarding pulmonary embolism (PE), a computerized tomography (CT) pulmonary angiogram was performed and showed a right lower lobe segmental PE, which was likely secondary to the bilateral popliteal deep vein thromboses in the lower extremities found later via ultrasound; he was started on heparin with eventual transition to apixaban. Two days post-admission, the sputum cultures returned positive for *Pseudomonas aeruginosa* and CRKP, though it was unclear whether either of these organisms were contributing to his illness given drastic improvement on the previously mentioned anticoagulants and antibiotic therapies. Due to his history of bilateral lung transplant and the ongoing use of immunosuppressing agents, the patient was started on polymyxin B as a prophylaxis with continued piperacillin/tazobactam; treatment with vancomycin was discontinued. Eleven days after admission, the patient’s vitals were stable, with no symptoms reported, and he was discharged on a 14-day course of ciprofloxacin.

### 3.7. Case 6 (2022)

A 73-year-old man with a history of hypertension presented with several months of progressive dyspnea and back pain. Three days prior to presentation he was hospitalized in Cambodia, where he had resided for the previous four years. There, he received IV antibiotics and was discharged with cefixime directly to the airport. He reported dyspnea for several months, which had become acutely worse prior to presentation, and he reported ambulating with a walker for the previous three months due to progressive back pain but denied focal neurologic deficits. He was originally from Central California and denied intravenous drug use and animal exposures.

On presentation, he was found to have acute bilateral pulmonary emboli and magnetic resonance imaging (MRI) changes, with concerns regarding thoracic vertebral osteomyelitis at T9-T10, acute T10 compression fracture, and an acute right 10th rib fracture. A biopsy of the vertebral lesion was terminated due to accidental entry into the intercostal artery. After the failed biopsy, vancomycin was started empirically. The antibiotics were changed to ertapenem after PCR testing of the blood showed *E. coli*, and he was discharged to a nursing facility with a plan for six weeks of therapy.

One week later he presented with altered mental status and was found to have COVID-19 and a loculated left pleural effusion. A repeat MRI noted decreased phlegmonous changes in the ventral space with progressive T9-10 destructive changes. When his mental status did not improve, concern was raised regarding carbapenem neurotoxicity, and the ertapenem was held. After four days, his mental status improved, and ceftriaxone and trimethoprim-sulfamethoxazole were started. A week later, he was transferred to the intensive care unit (ICU) with worsening agitation and delirium. The antibiotics were switched back to ertapenem and daptomycin. His clinical status worsened; he was intubated and required pressor support. His sputum culture grew CRKP, and his antibiotics were switched to cefiderocol. The cultures of the pleural fluid were negative. Over the next five days, he had a continuous fever and worsening multiorgan failure; ultimately, he died.

### 3.8. Genomic Characteristics

The multilocus sequence typing (MLST) analysis demonstrated genetic diversity within the CRKP isolates: ST-14, ST-16, ST-437, ST-2096, and ST-2497 (Table 2). There was no correlation observed between sequence type, specimen source, and disease. Moreover, different sequence types were found in patients with multiple CRKP isolates collected from a range of sources (Table 2).

A phylogenetic analysis was also performed to further analyze the genomic epidemiology of these CRKP (Figure 1A). The cgMLST-based phylogenetic tree showed the presence of two closely related clusters that were composed of either ST-14 (samples: 1A, 2A, and 3C) or ST-2497 (samples: 3A, 3B, 3D, and 5A). The ST-14 isolates were related to a strain found in Turkey in 2019 (NCBI Reference Sequence: NZ_CP094227.1), while the ST-2497 isolates were related to a strain found in Switzerland in 2020 (NCBI Reference Sequence: NZ_CP071086.1). An SNP analysis was conducted to further understand the genetic relatedness between the members of these two clusters. The ST-14 isolates had up to 36 SNP differences, whereas the ST-2497 isolates had up to 10 SNP differences (Figure 1B,C). Notably, the 2A and 3C isolates had 0 SNP differences, suggesting that they were genetically identical. These isolates were collected from two different patients in 2017 and 2018. However, no obvious epidemiological link between the two cases was identified.

We further compared these isolates with a few other representative single carbapenemase-producing CRKP isolated at our institution in the past 10 years, including three isolates producing NDM-1 (NDM Ctrl#1-3), two isolates producing KPC-3 (KPC Ctrl#1-2), and two isolates producing OXA-232 (OXA Ctrl#1-2). Figure 1A showed that there was no correlation between the carbapenemase genotype and the phylogenetic lineages, which was expected due to the active horizontal gene transfer being well known among *K. pneumoniae*, especially in the health care setting. Nevertheless, we did observe that ST-14, 16, and 2497 were the most frequently encountered CRKP genotypes in our institution.

### 3.9. AMR Profiles

A genomic analysis of the CRKP isolates revealed an extensive list of mutations, genes, and plasmids associated with AMR. All the CRKP strains were found to possess *pColKP3*-type plasmids and *Inc*-type plasmids, including *pIncFII* (Table 2). As expected, these isolates also carried plasmid-encoded carbapenemase genes such as *bla_OXA-181_*, *bla_OXA-232_*_,_
*bla_NDM-1_*, *bla_NDM-4_*, and *bla_NDM-5_*. Most (4/6) of the cases had CRKP co-producing NDM-1 and OXA-232. Case 1 had CRKP co-producing NDM-5 and OXA-232. Case 6 had CRKP co-producing NDM-4 and OXA-181. Interestingly, in Case 6 the patient had two CRKP isolates: one producing dual carbapenemase (NDM-1 and OXA-232) and another solely expressing OXA-232 (Table 2).

The other significant AMR genes identified included *bla_CTX-M-15_*, *armA*, *tet(A)*, and *tet(D)*, which conferred resistance to extended spectrum beta-lactams, aminoglycosides, and tetracyclines. These CRKP also possessed multiple mutations and AMR genes that confer additional antimicrobial resistance to sulphamethoxazole/trimethoprim, chloramphenicol, fluroquinolones, and macrolides (Table 2). Regardless of the differences in sequence type, these clinical isolates had a similar combination of antimicrobial resistance genes; consequently, they shared a comparable AMR profile and most likely shared similar mobile genetic elements.

## 4. Discussion

It is estimated that 4.95 million deaths are associated with AMR worldwide [30]. In particular, there has been an increase in the prevalence of CP-CRE due to the clonal spread and plasmid-mediated transmission of carbapenemase genes [31]. In 2008, NDM-1 was first identified in a patient who was treated for a urinary tract infection in India [32]. Initially discovered in a *K. pneumoniae* isolate, NDM-1 and its variants have now been found in other *Enterobacteriaceae* globally [33]. Moreover, the prevalence of OXA-48-like carbapenemases has increased significantly, in part due to the difficulty in its detection within the laboratory and, subsequently, the delayed pursuance of infection control [34]. In recent years, numerous cases have reported the high morbidity and mortality associated with CRKP co-producing dual carbapenemases worldwide [25,35,36,37,38,39,40].

On average, our laboratory has identified one CRKP co-producing NDM and OXA-181/232 case yearly since 2016. The source of infection for each patient was unclear. Notably, three patients in this study had previously traveled to South Asia, where NDM is known to be endemic [10]. All three patients had been hospitalized in these respective countries within one month of presentation to our facility, which suggests potential exposure and colonization during their stay. However, it should be noted that *Case 2′s* parotid mass had been present for several years prior to this hospitalization. All the patients in the cohort had a history of multiple hospitalizations prior to admission, which could represent opportunities for colonization as well. However, it is difficult to determine a definitive source given limited surveillance for CRE both within the US and abroad and the inherent limitations of our small cohort.

In our study, the diagnosis of CRKP during hospitalization was correlated with a high rate of mortality (>80%), though the patients’ demise was likely multifactorial in nature. Severe comorbidities were noted in four/five cases available for review, and this likely contributed to the poor patient outcomes. Additionally, several old generation antibiotics, such as polymyxin B and colistin, which were used to treat these patients, have since been found to have deleterious effects on kidney function and were found to worsen mortality in several randomized controlled trials [41]. Two of the four cases resulting in mortality were also found to have ongoing co-infections with both bacterial and fungal agents at the time of their death.

The treatment outcomes for all the patients, excluding *Case 5*, were uniformly poor despite variances in antibiotic therapies, infection source, time to diagnosis, and preexisting comorbidities. *Case 5′s* good outcome appears to have been due to their CRKP infection representing an incidental finding during treatment for COVID-19, which improved with treatment. Cases 2, 3, and 4 originally presented to our facility for the treatment of non-infectious issues (two were cardiogenic, one transplant-related), but ultimately suffered severe infectious complications that likely contributed to their ultimate demise. *Case 6′s* illness appears to have been a primary infection which eventually resulted in the patient’s death, though whether this was *E. coli*, *K. pneumoniae*, or another infection is unclear. Notably, 75% of the patients that received polymyxin B, which has since been dropped from the Infectious Diseases Society of America’s (IDSA) guidelines due to increased mortality and excess nephrotoxicity, had poor outcomes, with kidney dysfunction noted on the chart review; however, it is unclear whether poor renal function was related to the treatment or other ongoing comorbidities during their respective admissions [42].

The current guidelines suggest either ceftazidime-avibactam combined with aztreonam or cefiderocol alone for NDM-producing infections and ceftazidime-avibactam alone for OXA-48-producing infections outside of the urinary tract [42]. The treatment of infections within the urinary tract, which was not represented in our cohort, has been extensively discussed, and multiple agents are available on both an empiric basis and after susceptibility testing [42]. Given the concurrent production of NDM and OXA-181/232 in this cohort, treatment with either ceftazidime-avibactam and aztreonam or ceftazidime-avibactam and cefiderocol would have aligned best with the current guidelines [42]. However, these guidelines do not specify treatment in the case of parallel NDM and OXA-48-like production; this is likely due to the paucity of data on these organisms; so, the efficacy of treatment is unknown for these rare cases. For example, in our cohort, two (Cases 3 and 4) patients received what is now considered the most appropriate therapy (ceftazidime-avibactam and aztreonam), as well as the other adjunct agents (though it should be noted that treatment guidelines differed at the time of original patient presentation). Both individuals unfortunately died even with the appropriate therapy. However, these patients represented the two most ill patients in our cohort, having recently undergone cardiac arrest with ROSC and other ongoing complications that likely contributed to their ultimate demise, making it difficult to assess the efficacy of the antimicrobials in their cases. The current data and guidelines for the appropriate treatment of these NDM and OXA-48-like co-producers are not clear.

The genomic surveillance of these unusual CRKP can therefore provide critical information for better infection prevention in order to create treatment algorithms to help with patient management. Our results indicated that the most common CRKP in our cohort were the ST-14 isolates that co-produced NDM-1 and OXA-232; this has also been reported in several countries [43,44,45,46,47,48]. A phylogenetic relatedness analysis also revealed that the ST-14 isolates were closely related to a strain circulating in Europe and America. Moreover, the combination of carbapenemases identified in the other CRKP have also been reported worldwide [49,50].

All the isolates carried *pColKP3*-type and *Inc*-type plasmids, which are known to carry a range of carbapenemases and other AMR genes [15,51,52,53]. Additionally, *bla_CTX-M-15_* was identified in all the isolates. CTX-M-15 is the most predominant extended spectrum beta-lactamase (ESBL) that confers resistance to later generation cephalosporins [54]. Resistance to tetracyclines, which are broad-spectrum antibiotics, were also noted as these clinical isolates expressed either *tet*(*A*) or *tet*(*D*) [55]. Furthermore, most isolates expressed *armA* or *rmtB1*, which confer high-level resistance to aminoglycosides, including a newer drug, plazomycin [56].

An SNP analysis was performed to further assess the genetic relatedness among our isolates. There were 10 SNP differences among the ST-2497 isolates collected in 2018 and 2020, and only 36 SNP differences among the ST-14 isolates collected from 2016 to 2018, indicating that these CRKP strains were most likely circulating in a specific population sharing similar epidemiological exposures and risk factors (e.g., long-term ill patients residing in multiple health facilities). Given that the patients had differences in travel history and disease presentation, these data would suggest that some of these CRKP strains could be spreading in the Southern California community, most likely in hospitals and long-term care facilities.

The genetic surveillance of CRE co-producing dual carbapenemases is essential to monitor and control the spread within our community. Our results revealed that patients could harbor an assortment of CRKP genotypes with different carbapenemase(s) and AMR genes. By understanding the resistance mechanisms involved, we can provide critical information to enhance patient care and control the spread of infection. Our findings highlight the importance of the real-time genetic surveillance of CRKP co-producing NDM and OXA-48-like carbapenemases in Southern California.

## Figures and Tables

**Figure 1 microorganisms-11-01717-f001:**
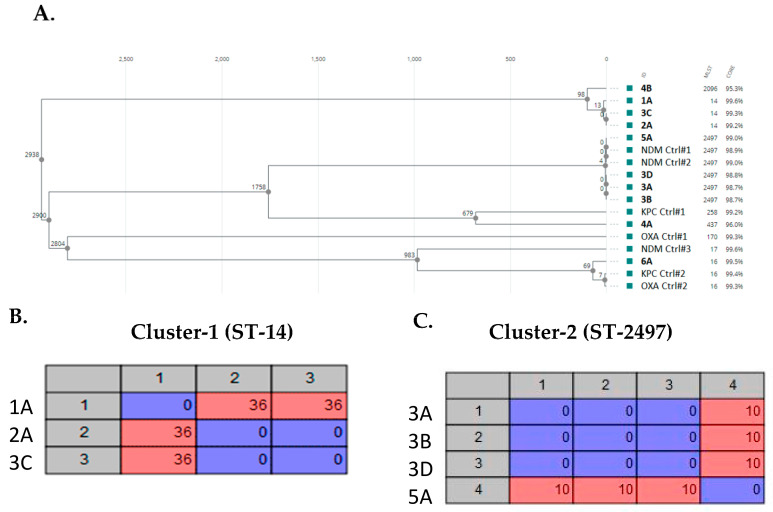
(**A**) Core genome multilocus sequence typing (cgMLST)-based phylogeny of the CRKP isolates. Nodes indicate clusters of related isolates. All samples were >95% fraction of core. SNP matrices of 2 closely related clusters of ST-14 (**B**) and ST-2497 (**C**).

**Table 1 microorganisms-11-01717-t001:** Summary of clinical cases associated with NDM + OXA-181/232 dual-carbapenemase-producing CRKP infections.

	Case 1	Case 2	Case 3	Case 4	Case 5	Case 6
Year	2016	2017	2018	2019	2020	2022
Age (years)	74	74	68	89	64	73
Clinical impression	Urinary tract infection	Parotid mass	Intra-abdominal abscess, sepsis	Cardiac arrest with traumatic spinal cord injury, sepsis	Hemoptysis, pneumonia	Weakness, dizziness
Antibiotics before CRE diagnosis	Unknown	Piperacillin-Tazobactam	None	Minocycline, Cefepime-Sulbactam, Tigecycline, Colistin, Metronidazole	Vancomycin, Piperacillin-Tazobactam	Ertapenem, Meropenem, Ceftriaxone, Trimethoprim-Sulfamethoxazole
Significant medical history	Unknown	Congestive heart failure, ischemic cardiomyopathy	Kidney transplant	Hypertension, Hypersensitivity lung disease	Lung transplant	Vertebrate osteomyelitis, atrial fibrillation
Specimen ID	1A	2A	3A, 3B, 3C, 3D	4A, 4B	5A	6A
Source	Urine	Parotid tissue	Peritoneal cavity fluid, hematoma, blood, abdominal tissue	Penile, blood	Expectorated sputum	Expectorated sputum
Travel	Unknown	Travel and hospitalized in India	No recent travel history	Travel to and hospitalized in India	No recent travel history	Lived in Cambodia for 4 years prior to infection
Treatment	Unknown	Meropenem, Polymyxin B	Aztreonam Cefiderocol, Ceftazidime-Avibactam, Plazomicin, Colistin, Polymyxin B, Tigecycline	Ceftazidime-Avibactam, Tigecycline, Piperacillin-Tazobactam, Meropenem, Polymyxin B, Aztreonam	Aztreonam, Polymyxin B	Cefiderocol
Outcome	Deceased	Deceased	Deceased	Deceased	Improved	Deceased

**Table 2 microorganisms-11-01717-t002:** MLST, plasmid types, and AMR profiles of CRKP clinical isolates.

	1A	2A	3A	3B	3C	3D	4A	4B	5A	6A
MLST	14	14	2497	2497	14	2497	437	2096	2497	16
Plasmids Types	ColKP3 IncFIA(Hl1) IncFIB(K) IncFII IncR	ColKP3 IncFIB(K) IncFII(K) IncR	Col(pHAD28) ColKP3 IncFIB(pQil)	Col(pHAD28) ColKP3 IncFIB(pQil)	ColKP3 IncFIB(K) IncFII(K) IncR	Col(pHAD28) ColKP3 IncFIB(pQil)	Col(BS512) ColKP3 IncFIB(pQil) IncFII(k)	ColKP3 ColRNAI IncFIB(K)	ColKP3 IncFIB(pQil)	Col4401 Col440ll ColKP3 IncFIB(pKPHS1) IncFII(K) IncX3
Aminoglycosides Resistance Genes	*aac(3)-lld* *aac(6′)-lb-cr* *ant(3″)-la* *aph(3″)-lb* *aph(3′)-Vl* *aph(6)-ld* *armA* *rmtB1* *sat2*	*aac(6′)-lb* *ant(3“)-la* *aph(3″)-lb* *aph(3′)-Vl* *aph(6)-ld* *armA* *sat2*	*aac(6′)-lb-cr* *ant(3″)-la* *aph(3′)-Vl* *armA*	*aac(6′)-lb-cr* *ant(3″)-la* *aph(3′)-VI* *armA*	*aac(6′)-lb* *ant(3“)-la* *aph(3″)-lb* *aph(3′)-VI* *aph(6)-ld* *armA* *sat2*	*aac(6′)-lb-cr* *ant(3″)-la* *aph(3′)-VI* *armA*	*aac(6′)-lb-cr* *ant(3″)-la* *aph(3′)-VI* *armA* *rmtF1* *rmtF2*	*aac(6′)-lb-cr* *ant(3″)-la* *armA* *sat2*	*aac(6′)-lb-cr* *ant(3″)-la* *aph(3′)-Vl* *armA*	*aac(6′)-lb-cr* *ant(3″)-la* *rmtB1*
Sulphamethoxazole/Trimethoprim Resistance Genes	*sul1* *sul2* *dfrA1* *dfrA12* *dfrA14*	*sul1* *sul2* *dfrA1* *dfrA12* *dfrA14*	*sul1* *dfrA14*	*sul1* *dfrA14*	*sul1* *sul2* *dfrA1* *dfrA12* *dfrA14*	*sul1* *dfrA14*	*sul1* *dfrA12* *dfrA14*	*sul1* *dfrA1* *dfrA12* *dfrA14*	*sul1* *dfrA14*	*sul1* *dfrA27*
Carbapenemase Genes	*NDM-5* *OXA-232*	*NDM-1* *OXA-232*	*NDM-1* *OXA-232*	*NDM-1* *OXA-232*	*NDM-1* *OXA-232*	*NDM-1* *OXA-232*	*NDM-1* *OXA-232*	*OXA-232*	*NDM-1* *OXA-232*	*NDM-4* *OXA-181*
Chloramphenicol Resistance Genes	*cmlA*	*catA1*	*catA1* *cmlA*	*catA1* *cmlA*	*catA1*	*catA1* *cmlA*	*catB* *floR2*	N.D.	*catA1* *cmlA*	N.D.
Chromosomal Beta-lactamase Genes	*SHV-11*	*SHV-28*	*SHV-11*	*SHV-11*	*SHV-28*	*SHV-11*	*SHV-11*	*SHV-28*	*SHV-11*	*SHV-1*
Plasmid-borne Narrow Spectrum Beta-lactamase Genes	*TEM-1* *OXA-1*	*TEM-1* *OXA-1*	*TEM-1* *OXA-1*	*TEM-1* *OXA-1*	*TEM-1* *OXA-1*	*TEM-1* *OXA-1*	*TEM-1* *OXA-1*	*TEM-1* *OXA-1*	*TEM-1*	*TEM-98*
ESBL Genes	*CTX-M-15*	*CTX-M-15*	*CTX-M-15*	*CTX-M-15*	*CTX-M-15*	*CTX-M-15*	*CTX-M-15*	*CTX-M-15*	*CTX-M-15*	*CTX-M-15*
Fosfomycin Resistance Genes	*fosA6*	*fosA6*	*fosA*	*fosA*	*fosA6*	*fosA*	*fosA*	*fosA6*	*fosA*	*fosA6*
Fluroquinolone Resistance Mutations	*gyrA*(D87G + *gyrA*(S83Y) + *parC*(S80I)	*gyrA*(D87G) + *gyrA*(S83Y) + *parC*(S80I)	*gyrA*(S83I) *+* *parC*(S80I)	*gyrA(S83l) +* *parC(S80l)*	*gyrA*(D87G) + *gyrA*(S83Y) + *parC*(S80I)	*gyrA*(S83I) *+* *parC*(S80I)	*gyrA*(S83I) *+* *parC*(S80I)	*gyrA*(D87G) + *gyrA*(S83Y) + *parC*(S80I)	*gyrA*(S83I) *+* *parC*(S80I)	*gyrA*(D87N) + *gyrA*(S83F) + *parC*(E84K)
Fluroquinolone Resistance Genes	*aac(6′)-lb-cr* *oqxA* *oqxB20* *qnrB1*	*oqxA* *oqxB20* *qnrB1*	*aac(6′)-lb-cr* *oqxA* *oqxB25* *qnrB1*	*aac(6′)-lb-cr* *oqxA* *oqxB25* *qnrB1*	*oqxA* *oqxB20* *qnrB1*	*oqxA* *oqxB25* *qnrB1*	*aac(6′)-lb-cr* *oqxA* *oqxB25* *qnrS1*	*aac(6′)-lb-cr* *oqxA* *oqxB20*	*oqxA* *oqxB25* *qnrB1*	*aac(6′)-lb-cr* *oqxA* *oqxB32* *qnrB6* *qnrS1*
Macrolide Resistance Genes	*ere(A)* *erm(B)* *mph(A)* *mph(E)* *msr(E)*	*mph(E)* *msr(E)*	*ere(A)* *mph(E)* *msr(E)*	*ere(A)* *mph(E)* *msr(E)*	*mph(E)* *msr(E)*	*ere(A)* *mph(E)* *msr(E)*	*mph(E)* *msr(E)*	*mph(E)* *msr(E)*	*ere(A)* *mph(E)* *msr(E)*	*mph(A)*
Multidrug Efflux Pump Genes	*emrD* *kdeA*	*emrD* *kdeA*	*emrD* *kdeA*	*emrD* *kdeA*	*emrD* *kdeA*	*emrD* *kdeA*	*emrD* *kdeA*	*emrD* *kdeA*	*emrD* *kdeA*	*emrD* *kdeA*
Tetracycline Resistance Genes	*tet(D)*	*tet(D)*	*tet(D)*	*tet(D)*	*tet(D)*	*tet(D)*	*tet(D)* *tet(G)*	*tet(D)*	*tet(D)*	*tet(A)*

N.D.: not detected.

## Data Availability

The sequence data have been deposited to NCBI Genbank (BioProject ID: PRJNA986585).

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
