# Peer review of "Clinical and Genomic Characterization of Carbapenem-Resistant Klebsiella pneumoniae with Concurrent Production of NDM and OXA-48-like Carbapenemases in Southern California, 2016–2022"

_microorganisms, 2023, doi:10.3390/microorganisms11071717_

Round 1

Reviewer 1 Report

The overall work is very well organized, perfectly written and executed. The study casts light on carbapenem-resistant Klebsiella pneumoniae, the spread of this resistance, the clinical outcome, pointing out to the need of more restrictive use of antibiotics. There are a few minor issues that the authors should address prior the publication of the manuscript, as follows:

1. Unfortunately, the manuscript was sent without line numbers, so it will be hard to point out to the lines that indicate the issues requiring correction. Nevertheless, Page 2, Line 2: Carbapenem-resistance should be corrected to “carbapenem-resistant”.

2. Page 2, Line 4-6: “carbapenemases producing” should be written as “carbapenemase-producing”, as well as in the caption of Table 1.

3. Page 6, Line 6: “Candida paraopsilosis” should be written correctly.

4. Page 8, the paragraph preceding Table 2: In this paragraph, the classes of antibiotics are mentioned such as fluoroquinolones, tetracyclines, chloramphenicol, macrolides. “Bactrim” is a trade name for sulphamethoxazole/trimethoprim and this trade name differs around the globe. It should be referred to as a class of antibiotics and not by its trade name. The same remark is applicable to Table 2 where “Bactrim” is mentioned.

5. Table 2: All the genes in Table 2 should be written in italic font style. The part citing the chromosomal mutations in the genes conferring resistance to fluoroquinolones is confusing and requires a better editing.

6. Table 2: The first column of the table indicates different parameters, where sometimes these parameters are genes and in other cases these are enzymes, in a bit confusing matter. It will be better represented when referring to genes (as in Aminoglycosides, Fosfomycin, macrolides, tetracycline, chloramphenicol) as (Aminoglycoside-encoding genes, Fosfomycin-encoding genes, etc., …) since the corresponding rows contain genes. Whereas the rows indicating “carbapenemases” or “ESBLs” or “chromosomal beta-lactamases” or ”narrow-spectrum beta-lactamases” will correspond to enzymes. Accordingly, genes and enzymes should be clearly differentiated to avoid confusion. The same remark goes to “efflux pumps”.

7. Discussion: Page 10, last paragraph, “Ceftazidime-avibactam” should not be written capitalized.

Author Response

Reviewer 1

The overall work is very well organized, perfectly written and executed. The study casts light on carbapenem-resistant Klebsiella pneumoniae, the spread of this resistance, the clinical outcome, pointing out to the need of more restrictive use of antibiotics. There are a few minor issues that the authors should address prior the publication of the manuscript, as follows:

  1. Unfortunately, the manuscript was sent without line numbers, so it will be hard to point out to the lines that indicate the issues requiring correction. Nevertheless, Page 2, Line 2:Carbapenem-resistance should be corrected to “carbapenem-resistant”.

Response: thank you for catching these errors; the line numbers are added and the typo is fixed.

  1. Page 2, Line 4-6: “carbapenemases producing” should be written as “carbapenemase-producing”, as well as in the caption of Table 1.

Response: thank you for catching this error; these typos are fixed.

  1. Page 6, Line 6: “Candida paraopsilosis” should be written correctly.

Response: thank you for catching this error; the typo is fixed.

  1. Page 8, the paragraph preceding Table 2: In this paragraph, the classes of antibiotics are mentioned such as fluoroquinolones, tetracyclines, chloramphenicol, macrolides. “Bactrim” is a trade name for sulphamethoxazole/trimethoprim and this trade name differs around the globe. It should be referred to as a class of antibiotics and not by its trade name. The same remark is applicable to Table 2 where “Bactrim” is mentioned.

Response: thank you for pointing this out; the drug name has been corrected.

  1. Table 2:All the genes in Table 2 should be written in italic font style. The part citing the chromosomal mutations in the genes conferring resistance to fluoroquinolones is confusing and requires a better editing.

Response: thank you very much for catching these errors; they have been fixed.

  1. Table 2:The first column of the table indicates different parameters, where sometimes these parameters are genes and in other cases these are enzymes, in a bit confusing matter. It will be better represented when referring to genes (as in Aminoglycosides, Fosfomycin, macrolides, tetracycline, chloramphenicol) as (Aminoglycoside-encoding genes, Fosfomycin-encoding genes, etc., …) since the corresponding rows contain genes. Whereas the rows indicating “carbapenemases” or “ESBLs” or “chromosomal beta-lactamases” or ”narrow-spectrum beta-lactamases” will correspond to enzymes. Accordingly, genes and enzymes should be clearly differentiated to avoid confusion. The same remark goes to “efflux pumps”.

Response: thank you for pointing out these inconsistencies; the names of these parameters have been standardized.

  1. Discussion: Page 10, last paragraph, “Ceftazidime-avibactam” should not be written capitalized.

Response: this is fixed.

Reviewer 2 Report

This article correlates the clinical and genomic characteristics of CRKP co-producing NDM and OXA-48-like carbapenemases isolated from patients in Southern California between 2016 to 2022.

Although this study is based on a very limited cohort, but it still provides a significant insight.

Line number is missing in the manuscript.

Have the authors looked into any retrospective study on CRKP from the hospital?

Could you provide more details about the parameters used in the single nucleotide polymorphism (SNP) phylogenetic analysis performed with the CLC Bio Genomic Workbench?

Why was only 1 isolate included for a few patients? Was there any difference in genotypic characteristics or different carbapenemases from other isolates?

Why did the authors decide 2 patients from outside the hospital setting?

What software did the authors use for assembling the reads?

Please cite a reference for 1928 Diagnostics.

The authors could include virulence gene characterization and other mobile genetic elements that contributes to the resistance in the study?

It would be better if the authors would include more CRKP isolates and a reference isolate for the phylogenetic tree to check the genetic relatedness.

No information provided on the data availability. Have the authors submitted the raw reads in any platform? It would be helpful to provide BioProject/GenBank accession numbers!

Good.

Author Response

Reviewer 2

This article correlates the clinical and genomic characteristics of CRKP co-producing NDM and OXA-48-like carbapenemases isolated from patients in Southern California between 2016 to 2022.

Although this study is based on a very limited cohort, but it still provides a significant insight.

Line number is missing in the manuscript.

Response: line number has been added.

Have the authors looked into any retrospective study on CRKP from the hospital?

Response: yes we have isolated numerous CRKP from our hospital in the past 10 years. We have added a few representative CRKP producing various single carbapenemase gene including KPC, NDM and OXA-48-like to show the overall genetic diversity of these CRKP (Figure 1A). We showed that except for the 2 clusters (ST-14 and ST-2497, Figure 1B&C) with close genetic relatedness, there was no correlation between the carbapenemase genotype and the phylogenetic lineages/cgMLST. This is expected due to the active horizontal gene transfer well known among K. pneumoniae especially in the health care setting. We have added a few sentences describing such findings (lines 296-303).

Could you provide more details about the parameters used in the single nucleotide polymorphism (SNP) phylogenetic analysis performed with the CLC Bio Genomic Workbench?

Response: we have added more details about how we performed the SNP analysis (lines 112 – 116).

Why was only 1 isolate included for a few patients? Was there any difference in genotypic characteristics or different carbapenemases from other isolates?

Response: we included all available isolates in this study, however, some patients only had 1 isolate saved for sequencing analysis. We have added other CRKP with different carbapenemase types into this study. See revised Figure 1A and lines 296-303.

Why did the authors decide 2 patients from outside the hospital setting?

Response: our laboratory also serves as a reference lab for higher-level antimicrobial susceptibility testing and sequencing analysis for clinical isolates with usual or discrepant drug susceptibility patterns. Therefore we included 2 patients from outside hospitals.

What software did the authors use for assembling the reads?

Response: we used CLC bio to assemble the reads. We have specified this in lines 116-117.

Please cite a reference for 1928 Diagnostics.

Response: thank you for this suggestion, we have cited 2 papers describing the bioinformatics pipeline developed by the 1928Dx (citation #28, 29).

The authors could include virulence gene characterization and other mobile genetic elements that contributes to the resistance in the study?

Response: thank you for this suggestion, however, we feel that virulence genes are not within the scope of this study, as we have shown that there’s no correlation between the carbapenemase genotype and the phylogenetic lineage. Therefore the discussion of virulence factors are not going to add any value to this paper. Our study focused more on the clinical impact of having these dual-carbapenemase-producing CRKP, and provided evidence of potential local transmission. Therefore, we didn’t intend to dive into the detailed genomic analysis. Regarding the mobile genetic elements, since usually these AMR genes are known to be carried on transposons that have been described else. Therefore, we also feel this discussion won’t add more value to this study.

It would be better if the authors would include more CRKP isolates and a reference isolate for the phylogenetic tree to check the genetic relatedness.

Response: we have added a few representative CRKP producing various single carbapenemase gene including KPC, NDM and OXA-48-like to show the overall genetic diversity of these CRKP (Figure 1A). We have added a few sentences describing these results (lines 296-303).

No information provided on the data availability. Have the authors submitted the raw reads in any platform? It would be helpful to provide BioProject/GenBank accession numbers!

Response: thank you for pointing this out. We have uploaded our sequence data to Genbank and indicated the project number (BioProject ID: PRJNA986585) (line 119).

Round 2

Reviewer 2 Report

The authors have addressed all the comments. The manuscript can be accepted in the current form.